# How do languages influence each other?
# Studying cross-lingual data sharing during LM fine-tuning

**Rochelle Choenni**
University of Amsterdam
r.m.v.k.choenni@uva.nl

**Dan Garrette**
Google Research
dhgarrette@google.com

**Ekaterina Shutova**
University of Amsterdam
e.shutova@uva.nl

## Abstract

Multilingual language models (MLMs) are jointly trained on data from many different languages such that representation of individual languages can benefit from other languages' data. Impressive performance in zero-shot cross-lingual transfer shows that these models are able to exploit this property. Yet, it remains unclear to what extent, and under which conditions, languages rely on each other's data. To answer this question, we use TracIn (Pruthi et al., 2020), a training data attribution (TDA) method, to retrieve training samples from multilingual data that are most influential for test predictions in a given language. This allows us to analyse cross-lingual sharing mechanisms of MLMs from a new perspective. While previous work studied cross-lingual sharing at the model parameter level, we present the first approach to study it at the data level. We find that MLMs rely on data from multiple languages during fine-tuning and this reliance increases as fine-tuning progresses. We further find that training samples from other languages can both reinforce and complement the knowledge acquired from data of the test language itself.

## 1 Introduction

Multilingual joint learning is often motivated by the idea that when multilingual language models (MLMs) learn information for multiple languages simultaneously, they can detect and leverage common universal patterns across them. Thus, these models can exploit data from one language to learn generalisations useful for another, obtaining impressive performance on zero-shot cross-lingual transfer for many languages (Wu and Dredze, 2019). Various studies suggest that representations created by popular MLMs, such as mBERT and XLM-R (Conneau et al., 2020), are not fully language-agnostic (Doddapaneni et al., 2021; Singh et al., 2019), but instead strike a balance between language-agnosticism and capturing the nuances of different languages through

language-neutral and language-specific components (Libovický et al., 2020; Gonen et al., 2020; Tanti et al., 2021). This naturally raises the question of how much models really benefit from multilingual data and cross-lingual sharing, and under what conditions this occurs. Many works have studied the encoding of cross-lingual patterns within MLMs by either focusing on probing for particular cross-linguistic differences (Ravishankar et al., 2019; Choenni and Shutova, 2022), or by analyzing the distributional properties of representational language subspaces (Yang et al., 2021; Rajaee and Pilehvar, 2022; Chang et al., 2022; Chi et al., 2020). Yet, it is not straightforward how to translate these results into model behavior at inference time. We aim to directly study how much influence languages exert cross-lingually on the predictions for individual languages.

In this study, we take a step back in the training pipeline to study the extent to which the model exploits its multilingual training data when making predictions. We hypothesise that if a model performs cross-lingual information sharing, then it will base its inference-time predictions (to some extent) on training data from multiple languages. Analyzing the cross-lingual sharing mechanism from the data reliance perspective leads to a set of interesting questions that we explore:

1. Given a test language $A$, does our MLM tend to base its predictions only on data from $A$ itself, or does it also employ data from other languages that it was exposed to during task fine-tuning?

2. Do MLMs only employ data cross-lingually out of necessity, e.g., in scenarios where in-language fine-tuning data is unavailable or insufficient?

3. Do languages support each other by adding similar information to what is relied upon from in-language data (i.e., reinforcing the

model in what it already learns), or do they (also) provide complementary information?

4. How do cross-lingual sharing dynamics change over the course of fine-tuning?

5. Is the cross-lingual sharing behaviour similar when the test language was seen during fine-tuning compared to when it is used in a zero-shot testing scenario?

To study this, we use TracIn (Pruthi et al., 2020), a training data attribution (TDA) method to identify a set of training samples that are most informative for a particular test prediction. The influence of a training sample $z_{train}$ on a test sample $z_{test}$ can be formalized as the change in loss that would be observed for $z_{test}$ if $z_{train}$ was omitted during training. Thus, it can be used as a measure of how influential $z_{train}$ is when solving the task for $z_{test}$.

To the best of our knowledge, we present the first approach to studying cross-lingual sharing at the data level by extending the use of a TDA method to the multilingual setting. We find that MLMs rely on data from multiple languages to a large extent, even when the test language was seen (or over-represented) during fine-tuning. This indicates that MLM representations might be more universal than previous work suggested (Singh et al., 2019), in part explaining the 'surprising' effectiveness of cross-lingual transfer (Pires et al., 2019; Wu and Dredze, 2019; Karthikeyan et al., 2020). Moreover, we find that cross-lingual sharing increases as fine-tuning progresses, and that languages can support one another by playing both reinforcing as well as complementary roles. Lastly, we find that the model exhibits different cross-lingual behaviour in the zero-shot testing setup compared to when the test language is seen during fine-tuning.

## 2 Background and related work

### 2.1 Training data attribution methods

TDA methods aim to explain a model's predictions in terms of the data samples that it was exposed to during training. Proposed methods include measuring the similarity between learned model representations from training and test samples (Rajani et al., 2020), and influence functions (Koh and Liang, 2017) that aim to compute changes in the model loss through Hessian-based approximations. While these methods compute influence between training samples and the final trained model, discrete

prediction-based methods like Simfluence (Guu et al., 2023) base influence on the full training trajectory instead. TracIn (Pruthi et al., 2020), used in this paper, is somewhere in between these methods: rather than using a direct loss difference, it tracks the similarity between gradients of training and test samples over model checkpoints. In NLP, TDA methods have so far mostly been used for unveiling data artifacts and explainability purposes (Han and Tsvetkov, 2022), for instance, to detect outlier data (Han et al., 2020), enable instance-specific data filtering (Lam et al., 2022), or to fix erroneous model predictions (Meng et al., 2020; Guo et al., 2021).

### 2.2 Studying cross-lingual sharing

Many approaches have been used to study the cross-lingual abilities of MLMs (Doddapaneni et al., 2021). Pires et al. (2019) and Karthikeyan et al. (2020) showed that MLMs share information cross-lingually by demonstrating that they can perform zero-shot cross-lingual transfer between languages without lexical overlap. This led to work on understanding *how* and *where* this sharing emerges.

One line of study focuses on how MLMs distribute their parameters across languages by analyzing the distributional properties of the resulting language representations. In particular, they aim to understand to what extent MLMs exploit universal language patterns for producing input representations in individual languages. As such, Singh et al. (2019) find that mBERT representations can be partitioned by language into subspaces, suggesting that little cross-lingual sharing had emerged. Yet others show that mBERT representations can be split into a language-specific component, and a language-neutral component that facilitates cross-lingual sharing (Libovický et al., 2020; Gonen et al., 2020; Muller et al., 2021). In addition, Chi et al. (2020) show that syntactic information is encoded within a shared syntactic subspace, suggesting that portions of the model are cross-lingually aligned. Similarly, Chang et al. (2022) more generally show that MLMs encode information along orthogonal language-sensitive and language-neutral axes.

While the previous works studied parameter sharing indirectly through latent model representation, Wang et al. (2020) explicitly test for the existence of language-specific and language-neutral parameters. They do so by employing a pruning method (Louizos et al., 2018) to determine the importance of model parameters across languages, and find that

some parameters are shared while others remain language-specific. Moreover, Wang et al. (2020) focused on the negative interference effects (Ruder, 2017) of cross-lingual sharing, i.e., parameter updates that help the model on one language, but harm its ability to handle another. They show that cross-lingual performance can be improved when parameters are more efficiently shared across languages, leading to new studies on finding language-specific and language-neutral subnetworks within MLMs to better understand (Foroutan et al., 2022) and guide (Lin et al., 2021; Choenni et al., 2022) cross-lingual sharing at the parameter level. In contrast to these works, we do not study cross-lingual sharing at the model parameter level, but instead consider sharing at the data level. To the best of our knowledge, we are the first to explore this direction.

## 3 Tasks and data

We conduct model fine-tuning experiments in three multilingual text classification tasks.

**Natural language inference (NLI)** The Cross-Lingual Natural Language Inference (XNLI) dataset (Conneau et al., 2018) contains premise-hypothesis pairs that are labeled with the relationship that holds between them: 'entailment', 'neutral' or 'contradiction'. The dataset contains parallel data in 15 languages. The original pairs come from English and were translated to the other languages. We use English, French, German, Russian and Spanish for model fine-tuning and testing.

**Paraphrasing** The Cross-Lingual Paraphrase Adversaries from Word Scrambling (PAWS-X) dataset (Yang et al., 2019) and task requires the model to determine whether two sentences are paraphrases of one another. To create this dataset, a subset of the PAWS development and test sets (Zhang et al., 2019) was translated from English to 6 other languages by professional translators, while the training data was automatically translated. We use English, French, German, Korean and Spanish.

**Sentiment analysis** The Multilingual Amazon Review Corpus (MARC) (Keung et al., 2020) contains Amazon reviews written by users in various languages. Each record in the dataset contains the review text and title, and a star rating. The corpus is balanced across 5 stars, so each star rating constitutes 20% of the reviews in each language. Note that this is a non-parallel dataset. We use Chinese, English, French, German and Spanish.

## 4 Methods

### 4.1 Models and fine-tuning

For all tasks, we add a classification head on top of the pretrained XLM-R base model (Conneau et al., 2020). The classifier is an MLP with one hidden layer and uses $\tanh$ activation. We feed the hidden representation corresponding to the beginning-of-sequence token for each input sequence to the classifier for prediction. We use learning rates of 2e-5, 9e-6, and 2e-5 for XNLI, PAWS-X, and MARC, and use AdamW (Loshchilov and Hutter, 2017) as optimizer. We fine-tune the full model on the concatenation of 2K samples from 5 different languages, i.e. 10K samples for each task. This allows us to limit the computational costs of computing influence scores (which increase linearly with the number of training samples), while still obtaining reasonable performance. We also reduce computational costs by converting each task into a simpler classification problem[1]: for XNLI, we follow Han et al. (2020) by classifying "entailment or not" (i.e., mapping neutral and contradiction samples to a *non-entailment* label); for MARC, we collapse 1 and 2 stars into a *negative* and 4 and 5 stars into a *positive* review category. We train for 10 epochs and use early stopping (patience=3). We find that training converges at epoch 4 for XNLI, and at epoch 5 for PAWS-X and MARC, obtaining 78%, 83%, and 90% accuracy on their development sets.

### 4.2 TracIn: Tracing Influence

We denote our training set $\mathcal{D} = \{z_i : (x_i, y_i)\}_{i=1}^{N}$, where each training sample $z_i$ consists of an input sequence $x_i$ and a label $y_i$. Koh and Liang (2017) show that we can compute how 'influential' each training sample $z_i \in \mathcal{D}$ is to the prediction for a test sample $x_{test} : \hat{y}_{test} = f_{\hat{\theta}}(x_{test})$. The influence score for a training sample $z_i$ on a test sample $z_{test}$ is defined as the change in loss on $z_{test}$ that would have incurred under the parameter estimates $f_{\hat{\theta}}$ if the model was trained on $\mathcal{D} \setminus z_i$. In practice, this is prohibitively expensive to compute as it requires training the model $|\mathcal{D}| + 1$ times. Koh and Liang (2017) show that we can approximate it by measuring the change in loss on $z_{test}$ when the loss

---

[1]This reduces computational costs indirectly because simpler tasks require fewer training samples to obtain reasonably high performance for. As a result, we need to compute influence scores between fewer training and test samples.

associated with $z_i$ is upweighted by some $\epsilon$ value:

$$
\begin{aligned}
\mathcal{I}(z_i, z_{test}) \approx \\
- \nabla_\theta \mathcal{L}(z_{test}, \hat{\theta})^T H_{\hat{\theta}}^{-1} \nabla_\theta \mathcal{L}(z_i, \hat{\theta})
\end{aligned} \quad (1)
$$

where $H_{\hat{\theta}}^{-1}$ is the inverse-Hessian of the loss $\mathcal{L}(\mathcal{D}, \hat{\theta})$ with respect to $\theta$, i.e. $[\nabla_\theta^2 \mathcal{L}(\mathcal{D}, \hat{\theta})]^{-1}$.

However, as computing $H_{\hat{\theta}}^{-1}$ is still expensive, this method requires further approximations if the model is non-convex, and they can be less accurate when used on deep learning models (Basu et al., 2021). Pruthi et al. (2020) find a similar, but first-order, solution that we use in this study, TracIn:

$$
\mathcal{I}(z_i, z_{test}) = \sum_{e=1}^{E} \nabla_\theta \mathcal{L}(z_{test}, \theta_e) \cdot \nabla_\theta \mathcal{L}(z_i, \theta_e) \quad (2)
$$

where $\theta_e$ is the checkpoint of the model at each training epoch. The intuition behind this is to approximate the total reduction in the test loss $\mathcal{L}(z_{test}, \theta)$ during the training process when the training sample $z_i$ is used. This gradient product method essentially drops the $H_{\hat{\theta}}^{-1}$ term and reduces the problem to the dot product between the gradients of the training and test point loss.

**Normalization** A problem of gradient products is that they can be dominated by outlier training samples of which the norm of their gradients is significantly larger than the rest of the training samples (Yu et al., 2020). This could lead TracIn to deem the same set of outlier samples as most influential to a large number of different test points (Han et al., 2020). In the multilingual setup, we know that dominating gradients is a common problem (Wang et al., 2020).[2] Barshan et al. (2020) propose a simple modification that we adapt: substituting the dot product with cosine similarity, thus normalizing by the norm of the training gradients.

## 5 Experimental setup

After fine-tuning our models (see Section 4.1), we in turns, use 25 test samples from each language for testing and compute influence scores between each test sample and all 10K training samples.[3] For each test sample, we then retrieve the top $k$ training

[2]From experiments using non-normalised FastIF (Guo et al., 2021), we found that outlier fine-tuning languages (e.g. Korean) would suspiciously often be ranked on top.

[3]Note that, despite using a NVIDIA A100 GPU, this is still computationally expensive because we have to recompute the influence scores between each test sample and all 10K training samples separately for each model checkpoint.

| $\mathcal{I}$ | Sentence pair |
|---|---|
| **test** | El río Tabaci es una vertiente del río Leurda en Rumania. |
| | El río Leurda es un afluente del río Tabaci en Rumania. |
| 4.19 | El río Borcut era un afluente del río Colnici en Rumania. |
| | El río Colnici es un afluente del río Borcut en Rumania. |
| 4.15 | El río Colnici es un afluente del río Borcut en Rumania. |
| | El río Borcut era un afluente del río Colnici en Rumania. |
| 4.10 | La rivière Slatina est un affluent de la rivière .. Roumanie |
| | La rivière Cochirleanca est un affluent de la .. Roumanie. |

Table 1: The top 3 most positively influential training samples retrieved for a Spanish test input from PAWS-X, and their corresponding influence scores ($\mathcal{I}$). Note that the last training samples are truncated to fit in the table.

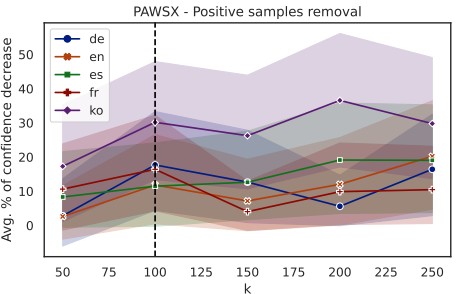

Figure 1: Average percentage of decrease in model confidence across test samples and fine-tuning languages when removing the top $k$ most positively influential training samples for PAWS-X.

samples with the highest influence scores and refer to them as the set of the most positively influential samples. Similarly, we refer to the top $k$ training samples with the most *negative* influence scores as the most negatively influential samples. Note that negative cosine similarity between gradients are commonly referred to as gradient conflicts (Yu et al., 2020) and have been shown to be indicative of negative interference in the multilingual setting (Wang et al., 2020; Choenni et al., 2022).

To pick the test samples for which we will compute influence scores, we select from the set of samples that the model labeled correctly, i.e. we study which training samples (and the languages they come from) positively and negatively influenced the model in making the correct prediction. For XNLI and PAWS-X, we train on parallel data; thus, as the content in our fine-tuning data is identical across languages, each language has equal opportunity to be retrieved amongst the most influential samples. Hence, we can ascribe the influence from each influential sample to the specific language that it is coming from as well as to the content of the sample itself (through the number of translations retrieved irrespective of the source language).

| ID | $\mathcal{I}$ | Sentence pair | P |
|---|---|---|---|
| **es test** | | Winarsky es miembro de IEEE, Phi Beta Kappa, ACM y Sigma Xi.
Winarsky es miembro de ACM, IEEE, Phi Beta Kappa y Sigma Xi. | + |
| de345 | 2.3 | Bernicat spricht neben Englisch auch Russisch, Hindi und Französisch.
Bernicat spricht neben Englisch auch Französisch, Hindi und Russisch. | + |
| en987 | 2.08 | The festival 's main partners are UBS , Manor , Heineken , Vaudoise Assurances and Parmigiani Fleurier.
The main partners of this festival are Parmigiani Fleurier , Manor , Heineken , Vaudoise and UBS . | + |
| fr987 | 2.04 | Les principaux partenaires du festival sont UBS, Manor, Heineken, Vaudoise Assurances et Parmigiani Fleurier.
Les principaux partenaires de ce festival sont Parmigiani Fleurier, Manor, Heineken, Vaudoise et UBS. | + |
| es115 | -2.16 | Il est le fils de Juan, a trois frères: Danilo Jr., Antonio, Danilo Rapadas et Cerila Rapadas ainsi que ses soeurs Roberta et Christina.
Il est le fils de Danilo Rapadas et de Cerila Rapadas. Il a trois frères, Danilo Jr., Antonio, Juan et ses soeurs Roberta et Christina. | − |
| ko115 | -2.13 | 그는 Juan의 아들이고 Danilo Jr., Antonio, Danilo Rapadas, Cerila Rapadas와 그의 아버지 Roberta와 Christina가 있습니다.
Danilo Rapadas와 Cerila Rapadas의 아들로 Danilo Jr., Antonio, Juan과 그의 자매 인 Roberta와 Christina가 있습니다. | − |
| es1771 | -2.06 | Además de Michael y Patrick, el álbum incluye contribuciones musicales de Diana, John, Chick, Stanley.
Además de Diana, el álbum contiene contribuciones musicales de Chick, Stanley, John, Michael y Patrick. | − |

Table 2: The top 3 most positively (top) and negatively (bottom) influential samples retrieved for a random test input from the PAWS-X dataset. $+$ indicates a correct paraphrase and $-$ an incorrect one. Also, correct re-ordered words are denoted by orange, incorrect ones by red and the respective words in the original sentence by green.

## 6 Quality of most influential samples

We qualitatively test the plausibility of our influence scores. In Table 1, we show a Spanish test input from PAWS-X and the corresponding top 3 most positively influential samples retrieved using TracIn. We see that TracIn ranks extremely similar samples with the same label as most influential. In Table 2, we also observe some evidence of cross-lingual sharing. The 3 most positively influential samples do not come from the test language, but they clearly test the model for the same knowledge: if the order of an unstructured list is slightly altered, do we get a correct paraphrase? In each case, this is correct. Yet, for the negatively influential samples, similar alterations are performed (i.e. changing the order of names), but in these cases this *does* crucially change the meaning of the sentences.

**Effect of $k$ on model confidence**    We run quantitative tests to assess the quality of our influence scores, and to select the optimal number for the top $k$ most influential samples to analyze in our further experiments. We hypothesize that only a subset of our fine-tuning data will substantially influence predictions, while a long tail of training samples will have little influence (either positively or negatively). To find this threshold value $k$, we select the top $k \in \{50, 100, 150, 200, 250\}$ most influential samples to test how our model confidence changes when leaving out these sets of samples from our fine-tuning data in turns. If our influence scores are meaningful, removing the top $k$ most positively influential samples will reduce the model confidence (i.e. the class probability) in the correct prediction, while removing the top $k$ most negatively influential samples should increase it. When we find the $k$ value for which the change in confidence con-verges, we conclude that the remaining samples do not exert much influence anymore, and we stop analyzing the ranking after this point.

**Results**    Figure 1 shows the effect of retraining the model on PAWS-X while removing the top $k$ most positively influential samples. We find that after $k$=100, the decrease in model confidence starts to level off. The same was found for negatively influential samples and XNLI. Thus, all further experiments focus on analysing the top 100 most influential samples (see Appendix A for more details on selecting $k$). Yet, while for XNLI removing the top 100 most positively influential results in a clear decrease in model confidence, removing the most negative ones does not result in a similar confidence increase. Thus, compared to PAWS-X, negative interference effects seem less strong in XNLI given our 5 fine-tuning languages. This is also reflected in Table 3 where we report the average influence scores between all training and test samples per language pair and task, and on average observe much higher scores for XNLI than for PAWS-X (see Appendix B for more statistics on the influence scores).

## 7 Cross-language influence

We now study how much each test language relies on fine-tuning data from other languages at test time for parallel and non-parallel datasets. Figure 2 shows, for all datasets, the percentage of training samples that contributed to the top 100 most positively and negatively influential samples based on their source language.

| | | Train | | | | | | Train | | | |
|---|---|---|---|---|---|---|---|---|---|---|---|
| | | de | en | es | fr | ru | | de | en | es | fr | ko |
| Test | de | .431 | **.442** | .425 | .434 | .418 | de | .244 | **.256** | .241 | .237 | .155 |
| | en | .633 | **.657** | .633 | .639 | .610 | en | .283 | **.308** | .285 | .279 | .153 |
| | es | .563 | **.603** | .597 | .587 | .542 | es | .221 | **.236** | .223 | .218 | .146 |
| | fr | .514 | **.540** | .525 | .529 | .499 | fr | .320 | **.335** | .325 | .323 | .189 |
| | ru | .651 | **.667** | .652 | .660 | .641 | ko | .143 | .146 | .141 | .140 | **.166** |

(a) XNLI      (b) PAWS-X

Table 3: For each language pair $(a, b)$, we compute the average influence score between all 2K training samples from the fine-tuning language $a$ and all the test samples from the test language $b$.

## 7.1 Parallel datasets

For XNLI and PAWS-X, across all test languages, the retrieved sets of most-influential training samples contain relatively high numbers of samples from languages other than the test language. This high degree of cross-language influence provides strong evidence of cross-lingual information sharing within the models. Korean (PAWS-X) is the only exception, which is least surprising as it is also least similar to the other fine-tuning languages and might therefore be processed by the model in relative isolation. Yet, we see that Korean still contributes cross-lingually to some extent (∼13% to the most positively influential samples on average). However, after further inspection we find that only in ∼11% of these Korean samples the sentences are fully written in the Hangul script; in all other cases, code-switching might be responsible for the cross-lingual alignment. Moreover, we observe that all test languages across both tasks mostly rely on data from their own language as most positively influential. Yet, the opposite does not hold: for instance, for PAWS-X we see that Korean is always the largest negative contributor irrespective of the test language, nicely showcasing the problem of negative interference (Ruder, 2017). Lastly, we find that while English obtains the highest average influence score across all training samples (see Table 3), this is not representative of its actual influence when judged by the most influential samples. This confirms our hypothesis that there is a long tail of training samples that are of little influence.

## 7.2 Non-parallel dataset

While parallel datasets allow for a fair comparison across languages in terms of the content that they were exposed to, this setting is not representative of most scenarios as most datasets are not parallel. Also, the translation of training samples across languages might decrease the variation between

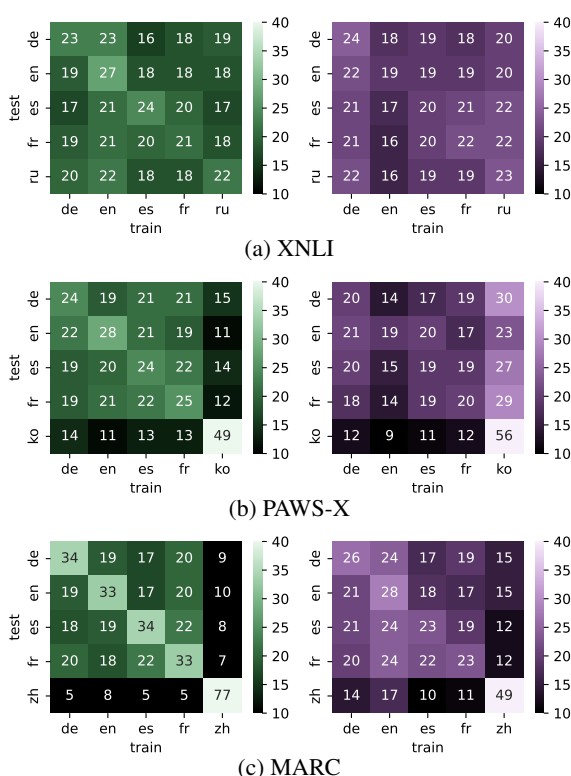

(a) XNLI

(b) PAWS-X

(c) MARC

Figure 2: For each test language, we show the percentage of samples that each fine-tuning language contributed to the top 100 most positively (left) and negatively (right) influential training samples averaged across all test samples.

languages, thus artificially boosting cross-lingual sharing within the models. Therefore, we also train a model on the non-parallel MARC dataset that contains user-written product reviews. In Figure 2c, we see that while languages indeed seem to rely more strongly on their own data for MARC compared to PAWS-X and XNLI (≈+10%), strong evidence for cross-lingual sharing is still observed. Moreover, similar language pair effects can be seen across tasks, e.g. French and Spanish rely on each other's data the most for both PAWS-X and MARC. Yet we also find interesting differences such as that for both parallel datasets, English contributes to the negatively influential samples the least, while for MARC it is instead the largest contributor. Given that our fine-tuning data is balanced across languages, it is possible that we are seeing the effect of translation here, i.e. parallel data is translated from English, which results in the other language data conforming more to English, a phenomena known as "translationese" (Koppel and Ordan, 2011). This is also supported by Table 3, where we found that on average the training samples from English obtained the highest influence scores, but for MARC

| Translations (%) | | de | en | es | fr | ko | ru |
|---|---|---|---|---|---|---|---|
| XNLI | POS | 60 | 59 | 58 | 62 | – | 60 |
| | NEG | 64 | 60 | 61 | 62 | – | 62 |
| PAWS-X | POS | 43 | 46 | 44 | 45 | 31 | – |
| | NEG | 45 | 50 | 46 | 46 | 32 | – |

Table 4: For the positively and negatively influential samples in the top 100 for each test language, we report how many of the samples coming from other fine-tuning languages are translations of the most influential samples from its own language (i.e. % reinforcing samples).

we find that Spanish most often obtains the highest scores instead (see Appendix B, Table 6).

# 8 Further analysis

We further analyze cross-lingual sharing for the tasks with parallel datasets since some of our analysis requires translation retrieval.

## 8.1 Complementary vs. reinforcing samples

Now that we have seen that our models rely on data from languages other than the test language, we study how these samples might contribute to the model performance, i.e., are they reinforcing the model with similar knowledge that it has seen from the test language, or do these samples somehow encode complementary knowledge that the model did not retrieve on from its own language? In order to make this distinction, we look at whether the most influential samples retrieved in other languages are translations of the most influential samples retrieved from the test language itself.

**Results** We report these percentages in Table 4, and find that for XNLI, over half of the contributions from different languages are translations of the most influential training samples from the respective test language, indicating that the model largely benefits from reinforcing data from other languages. For PAWS-X, this is not the case, indicating that here the biggest benefit of cross-lingual sharing can more likely be attributed to the model learning to pick up on new, complementary, information from other languages. As XNLI requires deep semantic understanding, we speculate that the model does not need to learn language-specific properties, but only needs to capture the content from data (possibly creating more universal representations to induce implicit data augmentation). Thus, the most influential samples might more often be translations since some samples are content-wise more influential, and samples across languages can contribute equally. Yet, for PAWS-X,

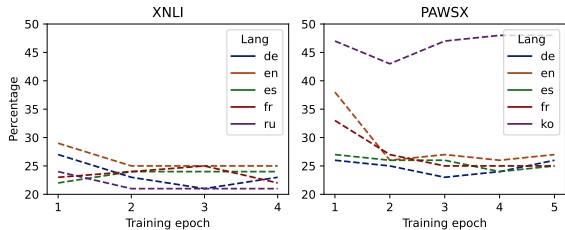

Figure 3: For each test language, we plot the percentage of samples coming from their own language that were included in the most positively influential training samples, i.e. the extent to which the model relies on its own language and how this changes over fine-tuning epochs.

the model requires some knowledge of grammatical structure, e.g. identical paraphrases can take different forms across languages, thus the model might learn from cross-lingual sharing differently.

## 8.2 Sharing dynamics during fine-tuning

As explained in Section 4.2, TracIn approximates influence over training, obtaining separate scores after each fine-tuning epoch. While in previous results we reported the sum of these scores, we now analyze them separately per fine-tuning epoch. Blevins et al. (2022) study cross-lingual pretraining dynamics of multilingual models to see when cross-lingual sharing emerges. We instead study whether different patterns emerge when looking at language influence across fine-tuning.

**Results** In Figure 3, we plot, for each test language, what percentage of samples from in-language data were included in the top 100 most influential samples across different fine-tuning epochs. From this, we see that for both tasks, the languages start relying less on their own fine-tuning data after epoch 2. Thus we conclude that on average the models gradually start to perform more cross-lingual sharing as fine-tuning progresses. Moreover, in line with previous findings (Blevins et al., 2022), we observe that the amount of cross-lingual sharing between different language-pairs fluctuates during fine-tuning (see Appendix C for results). To test whether the ranked influence scores between epochs are statistically significantly different, we apply the Wilcoxon signed-rank test (Wilcoxon, 1992), and confirm that between all epochs this holds true ($p$-value $< 0.05$).

## 8.3 Zero-shot testing

Another interesting testbed is zero-shot cross-lingual testing, in which no samples from the test

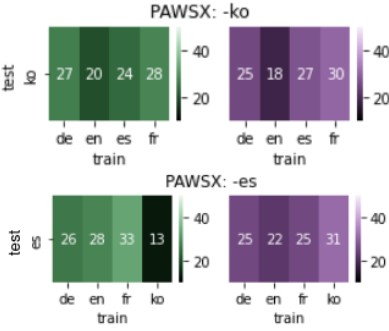

Figure 4: Percentage of samples that each fine-tuning language contributed to the top 100 most influential samples for Korean and Spanish during zero-shot testing.

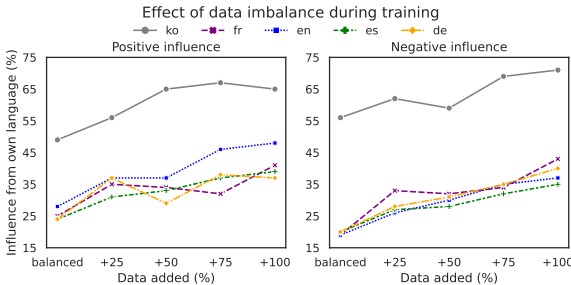

Figure 5: The percentage of data contributing to either the most positively (left) or negatively (right) influential samples for a particular language when adding $p$ % of data on top of that language's data during fine-tuning.

language are seen during fine-tuning; here, the model must rely solely on samples from other languages. Thus, for a language $l$, we compare the influential-samples ranking from the model for which $l$ was included in the fine-tuning languages $T$, $f_{\theta_T}$, to that of a model for which it was excluded, $f_{\theta_{T \setminus l}}$. We check for two potential differences: (1) Will $f_{\theta_{T \setminus l}}$ replace $l$-samples that were highly influential in $f_{\theta_T}$ with translations of those same samples?; and (2) Will $f_{\theta_{T \setminus l}}$ rely on the same non-$l$ samples that were highly influential in $f_{\theta_T}$?

As Korean was found to be the most isolated language for PAWS-X (i.e., it relies on data from other languages the least), while Spanish relies on cross-lingual sharing the most, we in turns retrain our model without Korean and Spanish, obtaining 74% and 81% accuracy respectively, and recompute the influence scores. We then compare the top 100 most influential samples from the two zero-shot models to those of $f_{\theta_T}$, and report how many translations of the samples from the test language vs. the other languages are covered.

**Results** Surprisingly, we find that, in the zero-shot setup, the models barely rely on the specific training samples that were found when the test language was included during fine-tuning. For Korean, only 5% of the most positively influential samples from the zero-shot model are direct translations of the Korean samples that were retrieved when it was included during training. Moreover, only 4% of training samples from the other languages that were retrieved were deemed most influential again in the zero-shot setup. The same trend was found for Spanish, albeit to a lesser extent, where translations of 14% of the Spanish and 13% from the other languages were recovered. Lastly, in Figure 4, we show the data reliance distribution across

fine-tuning languages for our zero-shot models. We find that the models still rely on cross-lingual sharing, and while Korean was previously processed in isolation (i.e., mostly relying on its own fine-tuning data), it now benefits from multiple languages.

## 8.4 Sharing as an effect of data-imbalance

An important aspect that can affect cross-lingual sharing is the (im)balance of language data during fine-tuning. For instance, if some languages are over-represented during training, then they might end up exerting stronger influence over other training languages, while languages that are under-represented might end up benefiting more from cross-lingual sharing (Wu and Dredze, 2020). To study how much the cross-lingual sharing effects observed so far can be ascribed to data scarcity, we perform experiments in which, for each language $l$ in turn, we fine-tune on PAWS-X, but with $p \in \{25, 50, 75, 100\}$% additional $l$ data included, thus ensuring that language $l$ is over-represented with respect to the other fine-tuning languages.

**Results** In Figure 5, we plot the percentage of in-language training samples that contribute to the set of most influential samples for the respective test language as an effect of data imbalance. For all languages, we see a clear trend: as the data gets more biased towards one language, training samples from that language increase in influence—both positive and negative—within the model. Yet we also see that this trend does not always steadily increase (e.g. for French and German). Moreover, for all languages except Korean, even when the language's data has fully doubled (+100%), its most influential samples set is still more than 50% from other fine-tuning languages. This indicates that even with data imbalances, the model largely ben-

efits from cross-lingual sharing. An interesting outlier is English, for which we see that positive influence from its own data rapidly increases (similar to Korean); we hypothesize that this could be due to being considerably overrepresented during pretraining, nudging the model towards processing this language in isolation as well.

## 9 Conclusion

To the best of our knowledge, we are the first to study the extent to which multilingual models rely on cross-lingual sharing at the data level. We show that languages largely influence one another cross-lingually, and that this holds under various conditions. Moreover, we find that cross-lingual sharing increases as fine-tuning progresses, and that languages can support one another both by playing a reinforcing as well as a complementary role. Lastly, we show how TracIn can be used to study data sharing in LMs. We hope that this paper can inspire future work on studying the sharing mechanism within multi-task and multi-modal models as well.

## 10 Limitations

One limitation of this study is that the experiments are computationally expensive to run, resulting in us only studying the effect on 125 test samples. Previous works have used more efficiency tricks to limit computational costs, for instance, by only computing influence scores between the test samples and the $k$ most similar training samples as found based on $k$ nearest neighbour search on their representations (Rajani et al., 2020; Guo et al., 2021; Jain et al., 2022). However, limiting the pool of training samples will bias us to retrieving samples based on the similarity between the hidden model representations from the final trained model. As one of our main goals is to study cross-lingual sharing from a new perspective, we opted against using such methods, and instead compute influence scores over the full training set.

Moreover, due to the computational costs, we are restricted to relatively easy tasks as (1) we can not use a large fine-tuning set and (2) TracIn operates on the sequence-level, i.e., it estimates how much a full training instance contributed to a prediction, making this method mostly suitable for classification and regression tasks. We suspect that cross-lingual sharing exhibits different cross-lingual behaviour for other types of tasks where language-specific information plays a bigger role at test time

(e.g. text generation or sequence labelling). In such tasks, the model could learn to rely on cross-lingual sharing to a lesser extent. Jain et al. (2022) recently extended influence functions to sequence tagging tasks to allow for more fine-grained analysis on the segment-level. Even though this further increases computational costs, it would be a good direction for future work on cross-lingual sharing.

Finally, we focus our analysis on the fine-tuning stage. However, pre-training and fine-tuning effects are hard to disentangle. As such, it would be reasonable to study the emergence of cross-lingual abilities during pre-training as well. However, in practice, this is difficult to achieve using TDA methods due to the large amount of training samples seen during pre-training. It requires (1) access to the full pre-training data, and (2) sufficient computational resources to compute influence scores between each test sample and (an informative subset of) training samples. Yet, given that many researchers are actively working on more efficient TDA methods, and our approach to studying cross-lingual data sharing can generally be coupled with any type of TDA method, we expect that similar studies that focus on the pre-training stage will be possible in the future as well.

## Acknowledgement

This project was in part supported by a Google PhD Fellowship for the first author. We want to thank Ian Tenney for his thorough feedback and insights.

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

# A Selecting $k$ for different tasks

Selecting a right threshold value for $k$ is not trivial as the number of most influential samples varies across languages and specific test samples. Moreover, in many cases, the top $k$ most positively influential training samples have the same label as the test instance, while the opposite holds true for the most negatively influential samples. Thus, when selecting a value for $k$ that is too large, we might not be able to distinguish between the effect of removing the most influential samples and the effect of data imbalance on our model. Thus, we opt for a more careful approach and select the smallest possible value of $k$ for which we observe consistent change in model confidence.

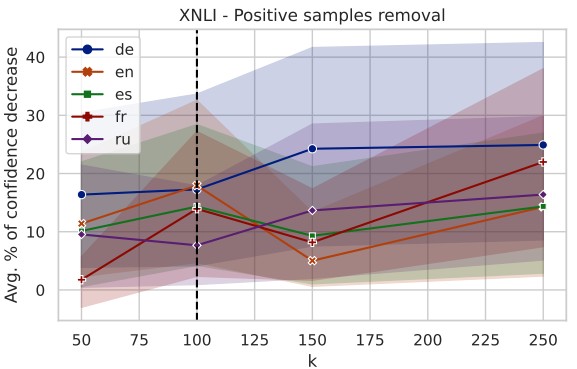

Figure 6: Average percentage (%) of decrease in model confidence across test samples and fine-tuning languages when removing the top $k$ most positively influential training samples for the XNLI dataset.

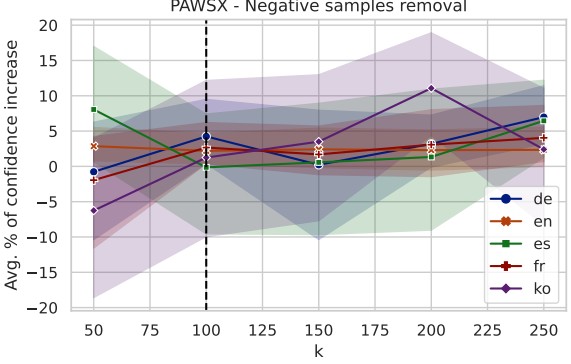

Figure 7: Average percentage (%) of increase in model confidence across test samples and fine-tuning languages when removing the top $k$ most negatively influential training samples from the PAWS-X dataset.

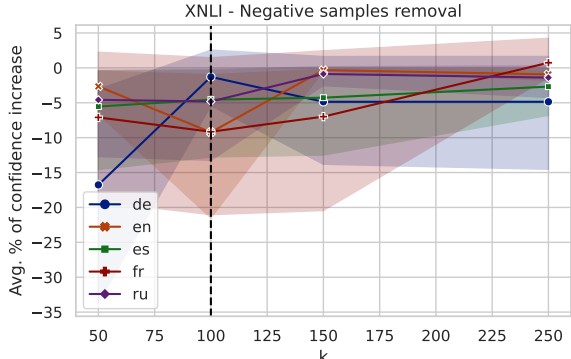

Figure 8: Average percentage (%) of decrease in model confidence across test samples and fine-tuning languages when removing the top $k$ most positively influential training samples for the XNLI dataset.

# B Influence score statistics

Figures 9, 10 and 11, show how for each task the influence scores between fine-tuning and test languages are distributed. We show separate plots for the distributions of positive and negative influence scores. In Table 5, we show an example of a random test input from XNLI and its corresponding top 3 most positively and negatively influential samples. In Table 6, we report average influence scores between training and test samples for MARC.

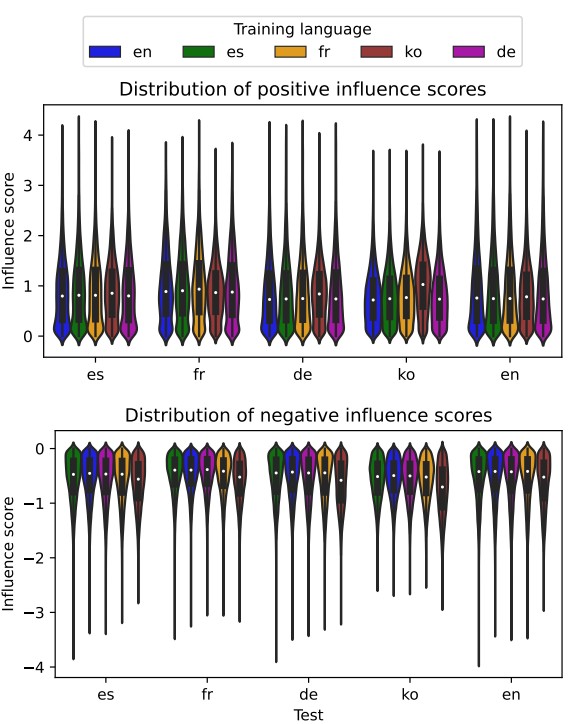

Figure 9: The distribution of influence scores for PAWS-X for all training samples from a language.

| ID / $\mathcal{I}$ | Premise and hypothesis | E |
|---|---|---|
| test | Ich bin mir also nicht wirklich sicher warum. 
 Ich bin mir bezüglich des Grundes sicher. | - |
| de935/2.40 | Und ich weiß nicht , was die Lösung ist. 
 Ich habe eine perfekte Vorstellung davon , was zu tun ist | - |
| en1696/2.34 | yeah i don't know why 
 I know why. | - |
| ru1696/2.30 | Да, я не знаю, почему. 
 Я знаю почему. | - |
| es758/-1.36 | Antes de la caída del comunismo, el Congreso aprobó sanciones amplias contra el régimen del apartheid en Sudáfrica. 
 El Congreso no apoyó el apartheid en Sudáfrica . | + |
| en1188/-1.33 | But there is one place where Will's journalism does seem to matter, where he does toss baseball. 
 Will's articles are only good in regards to sports | + |
| es1188/-1.14 | Pero hay un lugar donde el periodismo de will parece importar, donde él tira el béisbol. 
 Los artículos de will sólo son buenos en lo que se refiere a los deportes | + |

Table 5: An sample of the top 3 most positively (top) and negatively (bottom) influential samples retrieved for a random test input from the XNLI dataset. Note that + indicates a correct entailment and − a contradiction.

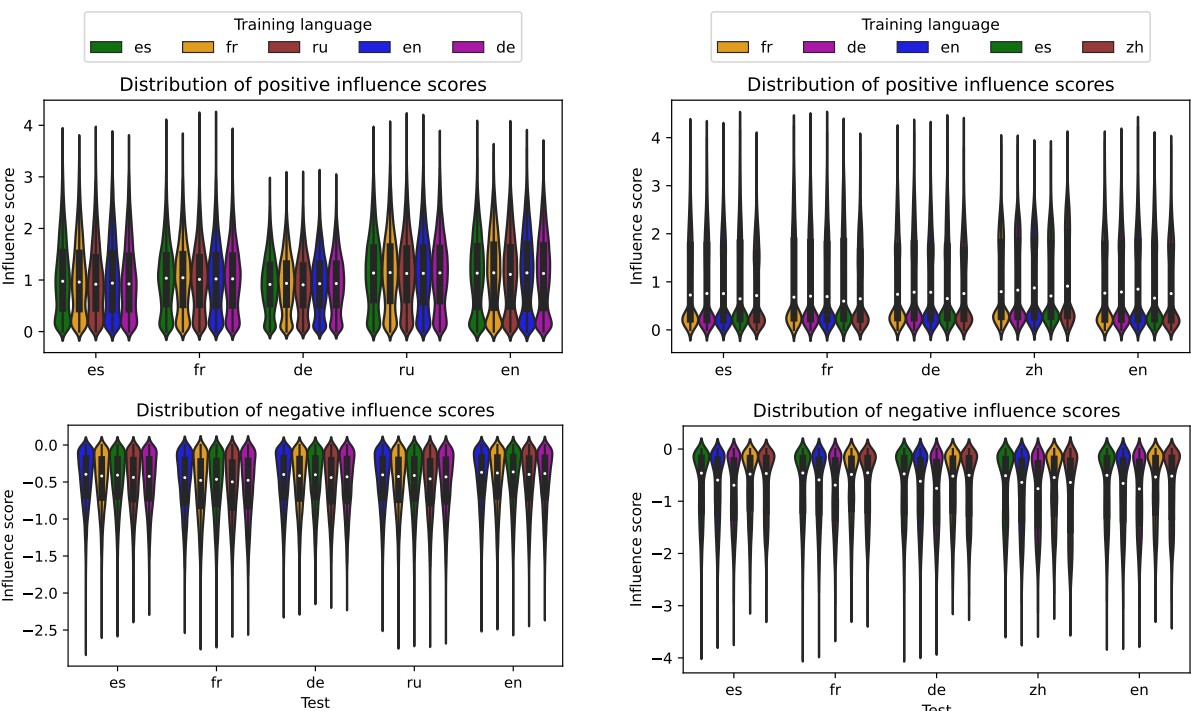

Figure 10: The distribution of influence scores for XNLI for all training samples from a language.

Figure 11: The distribution of influence scores for MARC for all training samples from a language.

## C Cross-language influence dynamics over fine-tuning epochs

In Figures 12 and 13, we show the full influence dynamics between all fine-tuning and test languages after different epochs during fine-tuning. Note that, to compare whether our ranked influence scores between different epochs are statistically significantly different, we applied the Wilcoxon signed-rank test (Wilcoxon, 1992), and we can confirm that between all fine-tuning epochs this holds true ($p$-value < 0.05).

|  | Train | | | | |
|---|---|---|---|---|---|
|  | de | en | es | fr | zh |
| de | .554 | .540 | **.589** | .582 | .455 |
| en | .540 | .554 | **.593** | .582 | .458 |
| Test  es | .539 | .536 | **.607** | .582 | .440 |
| fr | .561 | .556 | **.618** | .617 | .454 |
| zh | .535 | .544 | **.577** | .576 | .542 |

Table 6: For each language pair, we show the average influence score between all 2K training samples from a fine-tuning language and each test sample (from the respective test language) for the MARC dataset.

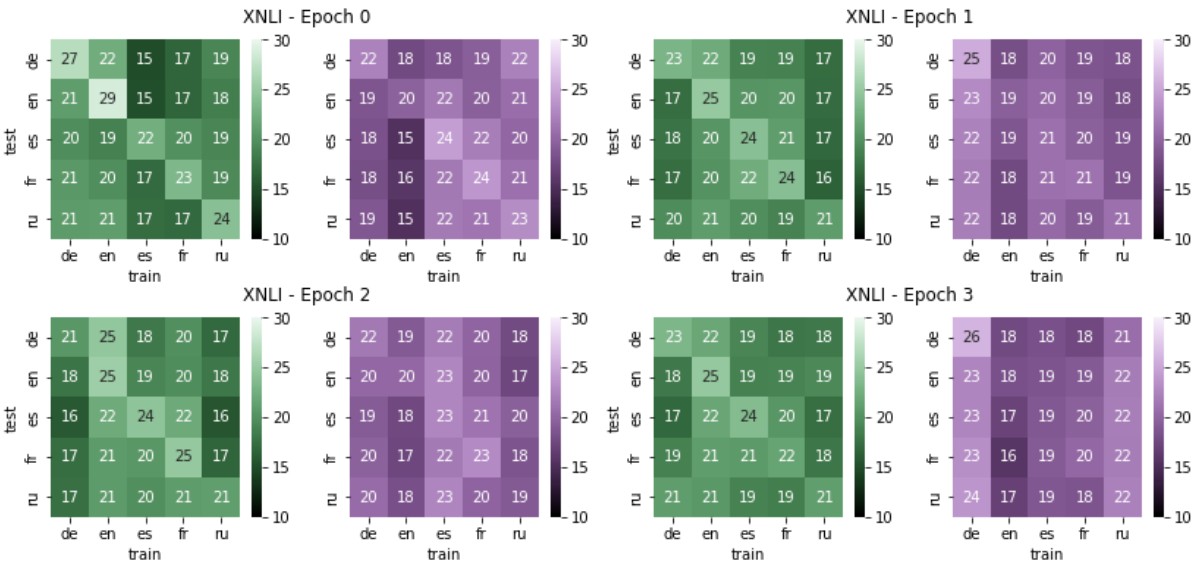

Figure 12: Full overview of how much each fine-tuning language exerts influence on each test language across the different fine-tuning epochs. We report percentages for which each fine-tuning language was represented in the test language's top 100 most positively (green) and negatively (purple) influential training samples.

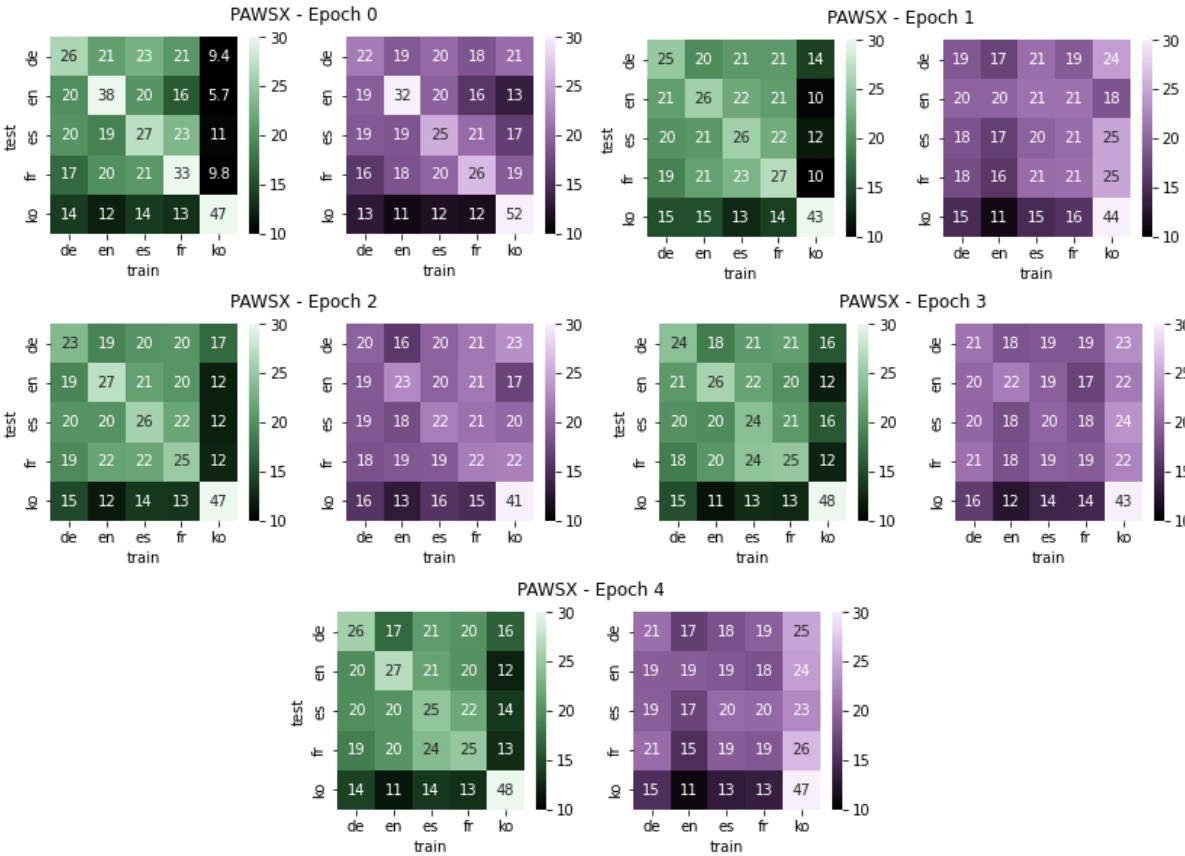

Figure 13: Full overview of how much each fine-tuning language exerts influence on each test language across the different fine-tuning epochs. We report percentages for which each fine-tuning language was represented in the test language's top 100 most positively (green) and negatively (purple) influential training samples.