# OpenReview forum: "How do languages influence each other? Studying cross-lingual data sharing during LM fine-tuning"
_EMNLP/2023/Conference — EMNLP 2023 Main_

### Official Review · Reviewer_qo5L · 2023-08-07

**Soundness:** 4

**Excitement:**

3: Ambivalent: It has merits (e.g., it reports state-of-the-art results, the idea is nice), but there are key weaknesses (e.g., it describes incremental work), and it can significantly benefit from another round of revision. However, I won't object to accepting it if my co-reviewers champion it.

**Paper Topic And Main Contributions:**

This paper studies cross-lingual sharing mechanisms during fine-tuning at the data level. They leverage a technique called TracIn the influence of each training language on test predictions in a given language. This sheds light on very important aspects of cross-lingual data-sharing dynamics including to what extent languages influence each other, complementarity and reinforcement nature of the interactions between languages, and how all of that plays out in data-scarce and zero-shot scenarios.

**Reasons To Accept:**

- The first data approach to studying cross-lingual sharing at the data level.
- This data-level study of cross-lingual sharing is very important to the multilingual NLP community as it will help design MLLMs in such a way as to maximize cross-lingual sharing by understanding the underlying factors that encourage or limit that. It could also be directly used as a measure of the intrinsic performance and cross-lingual capabilities of those models.
- The quality of the approach chosen allows an unbiased measuring of the influence over fine-tuning data. I like how the paper chooses quality over quantity (as explained in the limitations).

**Reasons To Reject:**

- Shedding light on the mechanism of cross-lingual sharing in MLLMs should involve more studying their biases at the pre-training stage and incorporating an understanding or measure of those to assess the true influence of languages at the fine-tuning stage. I think without that the study is a bit lacking and inconclusive in that.
- While this is the first study of cross-lingual sharing at the data level, it could help to compare those insights to the parameter level to see how complementary or orthogonal they are.
- While I think the ideas are clear, the paper could benefit from another round of restructuring especially when it comes to making a distinction between the methodology and experimental setup (check comments below).

**Reproducibility:**

4: Could mostly reproduce the results, but there may be some variation because of sample variance or minor variations in their interpretation of the protocol or method.

**Reviewer Confidence:**

3: Pretty sure, but there's a chance I missed something. Although I have a good feel for this area in general, I did not carefully check the paper's details, e.g., the math, experimental design, or novelty.

**Typos Grammar Style And Presentation Improvements:**

- Section 3: this should go into the experimental setup
- Section 4.1: some parts of it should also go into the experimental setup.
- You could start by Section 4.2 which is the main approach of the paper then leave describe just the methodology of different evaluation tasks after that in the methods section. After that, you can talk about the different hyperparameters of those tasks and stats of datasets in the experimental setup. You basically want to decouple your approach for studying/quantifying the influence of languages from the downstream tasks that are used as use cases for this analysis and make your approach look as generic enough to be used to understand any downstream task that the NLP community is willing to study next.
- Table 1: what the first column? explain in the caption.
- Line 377 Results: could be merged with the previous paragraph
- Section 7: you should explain here that the next subsections will talk about two types of datasets all in Figure 2
- Table 3: table not well introduced or explained in main text

---

> ### Author Rebuttal · Authors · 2023-08-28
>
> Thank you very much for the positive review.
>
> While pre-training definitely affects the behavior of the model at the fine-tuning stage, studying the effects of different pre-training runs on the fine-tuning behavior would be a different study. Studying the pre-training phase can not directly tell us about the predictions that are made at test time as: (1) we typically need to first fine-tune the model on a downstream task and (2) during fine-tuning the way in which the model learns can change. Moreover, studying the fine-tuning stage allows us to compare cross-lingual data sharing for different downstream tasks and datasets. This allowed us to pose a set of interesting questions about which types of tasks elicit more sharing and the effect that the dataset properties could have on this (e.g. parallel vs non-parallel data).  Thus, given the wide range of experiments reported on in the paper already, studying the pre-training effects was left for future work.
>
> Moreover, while we do not perform experiments at the parameter level ourselves, we do compare our findings to those obtained from previous studies at the parameter level. For instance, in line with previous studies at the parameter level, we find that cross-lingual sharing increases during fine-tuning and that the amount of sharing between language pairs fluctuate over training. While running more experiments at the parameter level ourselves would be interesting, we believe that the set of experiments we picked to conduct at the data-level already tells an interesting, complete, story on its own. Also, since we are limited to 8 pages, we do not think that it would be feasible to add more experiments to the paper. Though, we do believe that directly contrasting results at the parameter and data level could result in an interesting follow up paper.
>
> Lastly, thank you for the feedback on the structure of the methods and experiments section. We will incorporate all of your suggestions in the camera-ready version.

---

### Official Review · Reviewer_3fXq · 2023-08-07

**Soundness:** 4

**Excitement:**

4: Strong: This paper deepens the understanding of some phenomenon or lowers the barriers to an existing research direction.

**Paper Topic And Main Contributions:**

Many of the previous works studying multilingual capabilities of LLM have proposed to look at cross-lingual capabilities by probing or by analyzing distributional properties of language subspace. In this work authors propose to take a step back in the training pipeline and study in which extent the model exploits its multilingual training data to make cross-lingual predictions.

Authors use the TracIn method previously published, which allows to identify the most informative training samples. The tasks are fairly standard for the topic (XNLI, cross-lingual paraphrasing, cross-lingual Sentiment Analysis), choice that helps for comparison with prior works. Results presented are interesting and highlight that LM cross-lingual representations may be more universal that what previous works suggested.

**Questions For The Authors:**


Dear Authors, thank you for the work. Please find below a few questions:
 * L248: How making it a binary classification helps with the computational cost? Arguably making it a multi-classification task is not more expensive. More generally, it seems most of experiments in this paper rely on XLM-R, which with its 550M parameters can easily be run on a modern single MacBook Pro/comparable laptop. L311: Authors report using an A100 GPU. Can you elaborate more about the 'very expensive' cost?
* Table 1: Why the last sample in Table 1 has '..'. Does that indicate a truncation or is that the sample?
* Do you plan to release the code used for the experiments? All the models and datasets are open-source. Releasing the code would allow the community to reproduce/replicate the results presented in this work.

**Reasons To Accept:**

 * Paper is well written and presented. All questions and findings are presented in Section 1. Formal notations are correctly introduced in subsequent sections, and they are consistent between each other.
 * The work presented seems to be the first to use a training data-first approach to study cross-lingual aspects. Previous lines of work have studied cross-lingual abilities of LM at the parameters or prediction/probing level.
 * Experiments are clear and well documented. Additional experiments are added in Section 8 and details in Appendix. Great point as well to note: many examples are given and manually analyzed.
 * Results presented are interesting for the community as they highlight that LM cross-lingual representations may be more universal that what previous works suggested.

Overall this paper is well written and experiments are correctly conducted and presented. Results are interesting and could interest as well the rest of the research community.

**Reasons To Reject:**

* Multiple times the authors mention the expensive cost related to their experiments. However, looking at the models involved (XLM-R with 550M parameters), this seems far away from today standards (> 4B parameters, with 'small' models like FlanT5-xl). This raises some concerns:
    (a) It is not extremely clear in the paper why this computational cost seems so expensive. Adding measurements (e.g. estimated FLOP per experiment) would have helped grasping the problem.
    (b) If the cost is indeed expensive, this seems to limit importantly the scope of this work as model like XML-R are fairly small.

**Reproducibility:**

3: Could reproduce the results with some difficulty. The settings of parameters are underspecified or subjectively determined; the training/evaluation data are not widely available.

**Reviewer Confidence:**

3: Pretty sure, but there's a chance I missed something. Although I have a good feel for this area in general, I did not carefully check the paper's details, e.g., the math, experimental design, or novelty.

**Typos Grammar Style And Presentation Improvements:**

* L339: A enumerated list seems to be introduced in L339 with a 'First', however the second item seems to be not present. If this is not a list of findings, I suggest to remove "First".
* Figure 1: k starts at 50. Is that intended?
* Figure 4 is not readable. Could you vectorize it like the other pictures/figures?

---

> ### Author Rebuttal · Authors · 2023-08-28
>
> Thank you very much for your review and appreciation of our paper.
>
> The problem with the compute expense is that we need to get the gradients of the loss for each training and test sample to compute the dot product between them. Though it is not possible to simply store the gradients of ~10K training samples into memory. Therefore, we need to re-compute the gradient for each training sample for every test sample. Moreover, note that you have to do this for each model checkpoint (in our case we use 4-5 checkpoints). Also, computing the dot products between such high dimensional vectors is expensive. We would like to point out, however, that while we focused on retrieving high quality influence scores on the full training set for this first explorative study (such that we wouldn’t miss any training sample or bias the results), researchers are in parallel actively working on various efficiency tricks that can be used to speed up the process. Given that our approach to studying cross-lingual data sharing can be coupled with any type of TDA method, we hope that many interesting follow up experiments will be possible using faster methods as well.
>
> RE the binary classification: It reduces computational costs indirectly because it simplifies the task and thus requires fewer training samples (only 10K) to obtain reasonably high performance on the task. Therefore we need to compute influence scores between fewer training and test samples which reduces computational costs. We will clarify this in the paper.
>
> RE table 1: Yes, this is just a truncation to make the example fit in the Table. We will clarify this in the caption.
>
> RE the code: Yes, we will release our code upon publication.
>
> Typos Grammar Style And Presentation Improvements:
> - L339: A enumerated list seems to be introduced in L339 with a 'First', however the second item seems to be not present. If this is not a list of findings, I suggest to remove "First".
>
> We will remove it.
>
> - Figure 1: k starts at 50. Is that intended?
>
> Yes, we started by trying to remove the top 50 most influential samples.
>
> - Figure 4 is not readable. Could you vectorize it like the other pictures/figures?
>
> Thank you for pointing this out, we will make sure to improve the readability in the camera-ready version.

---

### Official Review · Reviewer_JCHa · 2023-08-10

**Paper Topic And Main Contributions:** 1. This paper attempts to explore the…
**Soundness:** 4

**Excitement:**

4: Strong: This paper deepens the understanding of some phenomenon or lowers the barriers to an existing research direction.

**Questions For The Authors:**

I'm more curious if this conclusion can be generalized for the pre-training phase of MLLMs? Or can it only be limited to the fine-tuning stage, as the authors say?

**Reasons To Accept:**

1. This paper attempts to explore the question of how much models really benefit from multilingual data and cross-lingual sharing, and under what conditions this occurs. The question explored in this paper is very interesting and important, and while some works have tried to explore this question and given some conclusions, this work presents a new perspective to analyze the question. While it is not difficult to think of analyzing this question from a data perspective, it is nice that this work succeeds in implementing this idea.
2. This paper hypothesizes that if a model performs cross-lingual information sharing, then it will base its inference-time predictions (to some extent) on training data from multiple languages. This hypothesis appears to be correct, while the authors conduct reasonable experiments based on this hypothesis.
3. This paper is well-written and clear.

**Reasons To Reject:**

1. My only point of concern is that this paper only conducts experiments on one MLLM, the XLM-R base. For the question the paper wants to explore, will different MLLMs or different model size behave differently?
2. It would not generally be considered that the XLM-R base is an LLM, which seems a bit of an overstatement. Also, I think the authors should add other types of PLMs to make their statement more strong, such as decoder-only PLMs

**Reproducibility:**

3: Could reproduce the results with some difficulty. The settings of parameters are underspecified or subjectively determined; the training/evaluation data are not widely available.

**Reviewer Confidence:**

3: Pretty sure, but there's a chance I missed something. Although I have a good feel for this area in general, I did not carefully check the paper's details, e.g., the math, experimental design, or novelty.

---

> ### Author Rebuttal · Authors · 2023-08-28
>
> Thank you very much for the positive review.
>
> While it is possible that models with e.g. larger sizes would behave differently, we believe that it is already an interesting finding that at least one popular LM can rely on cross-lingual data sharing. It would be unlikely that better and larger models would not have the ability to learn this as well. Though, what we could instead find is that larger models do not need to do it because of their increased capacity. As this was outside the scope of this project, we hope that the approach proposed in this paper will enable such follow up studies.
>
> Also, given the recent advancements in the field, we will change our use of large language models to just language models.
>
> Lastly, while running experiments to study the effect during pre-training would be very interesting, the problem with this would be that at the pre-training phase too many training samples are seen to  compute influence scores for. We would need to re-run pre-training to make sure that we know what exact training samples were seen and compute influence scores between each training and test sample (or an informative subset of training samples – though even computing scores for just e.g. 1% of the pre-training data will already be expensive). We can imagine that other research groups will have the compute power to do something like that in the future though, especially since researchers are in parallel actively working on more efficient TDA methods, and our approach to studying cross-lingual data sharing can generally be coupled with any type of TDA method.

---

### Official Review · Reviewer_FsHG · 2023-08-10

**Soundness:** 4

**Excitement:**

3: Ambivalent: It has merits (e.g., it reports state-of-the-art results, the idea is nice), but there are key weaknesses (e.g., it describes incremental work), and it can significantly benefit from another round of revision. However, I won't object to accepting it if my co-reviewers champion it.

**Paper Topic And Main Contributions:**

The paper analyses how multilingual training data is used by a XLM-R model specifically fine-tuned for binary classification tasks. The authors use a Training Data Attribution (TDA) method to understand which samples of a training corpus are more influential when making predictions in given languages. The authors explore a big variety of experiments. They start analysing the differences between using a parallel corpus across languages, or a non-parallel one. Using the parallel dataset they analyse if data of different languages reinforce or complement each other in a prediction. They also analyse how the language-sharing varies during the training, what happens in zero-shot conditions and with an unbalanced training set. The contribution of the paper is gaining insights on how the multilingual data is leveraged in the stated settings. In general, the authors observe that the model frequently relies on data from different languages, and not only in the given one for the prediction. They see that the sharing increases as the fine-tuning progresses and that the shared information both reinforces and complements the data of the test language.

**Reasons To Accept:**

- The paper is well written, with figures and tables that enhance comprehension.
- The experiments section is very well done. There is a big variety of experiments covering all possible settings. Every time I was thinking about an experiment that would be interesting to try, the authors had already included it in one of the following sections.
- The authors explore different datasets and variety of languages, including different alphabets.
- The conclusions drawn after each experiment are clear and well supported by the experiments.

**Reasons To Reject:**

Affected my rating:
- The model used is a XLM-R model (Conneau et al., 2020), which I consider old in the field of LMs considering recent advancements.
- The title and abstract of the paper state the analysis is done on a *Large* Language Model, which again considering recent advancements, is not how I would define a XLM-R model. I understand that the TDA methods used are very computationally expensive, and therefore it would be hard to apply them on bigger models because of potential resources limitations. However, then the paper should not talk about *Large* Language Models, and just about Language Models instead.

Did not affect my rating:
- The authors explore just one model, therefore it is not clear if this results would extend to other language models. This is understandable due to the resources needed to obtain the attributions.
- Limited impact. It is an interesting exploratory analysis, but some previous work had obtained similar conclusions and I don't see how the results could be used for further research.

**Reproducibility:**

5: Could easily reproduce the results.

**Reviewer Confidence:**

3: Pretty sure, but there's a chance I missed something. Although I have a good feel for this area in general, I did not carefully check the paper's details, e.g., the math, experimental design, or novelty.

---

> ### Author Rebuttal · Authors · 2023-08-28
>
> Thank you very much for the positive review.
>
> While we acknowledge that larger models have come out since XLM-R, it was useful for this work to study XLM-R specifically as this model’s cross-lingual abilities has been widely studied at the parameter level in previous studies. Analyzing a different model wouldn’t allow us to directly contrast our findings to existing knowledge about multilingual models.  Moreover, we believe that studying smaller models like XLM-R can still yield insightful results that can guide design decisions for larger ones as well (especially since they all still largely rely on the same training strategies and the transformer architecture). Though, given the recent advancements in NLP, we agree that it might be more suited now to call XLM-R a language model instead of a large language model. We will adjust this accordingly.
>
> Also, while it is true that we can not directly generalize the results from XLM-R to other LMs, we would like to point out that the fact that XLM-R is successfully able to perform cross-lingual data sharing, makes it very unlikely that larger, stronger, LLMs would not have the ability to learn this property as well. Whether it also uses it would be a question for future work.
>
> Lastly, to better clarify the usefulness of this study: We believe that interpretability studies like these will help us understand the current limitations of multilingual models. In turn, this can help us guide design decisions and future research directions. Had we, for instance, found that cross-lingual sharing does not already happen at the data level, it would make sense to focus future efforts on forcing  the model to rely more on cross-lingual data sharing in the hope to further improve performance. Also, as Reviewer qo5L nicely pointed out, we could use this approach as an intrinsic measure for testing the cross-lingual abilities of LMs. Moreover, we could, for instance, test to what extent cross-lingual data sharing correlates with better downstream task performance, i.e. testing to what extent cross-lingual data sharing actually benefits performance. In short, we believe that this approach opens up the possibility of exploring many new research questions.

---

### Meta-Review · Area_Chair_BnGG · 2023-09-19

**Recommendation:** 5

**Metareview:**

In the context of multilingual masked language models, this paper focuses on understanding the importance of specific training (i.e. fine-tuning) data samples at test time (referred to as cross-lingual data sharing). More specifically, it measures and analyzes the importance of the training samples and their language distribution using TracIn (Prudi et al. 2020) for the multilingual and cross-lingual performance of XLM-R across three tasks.

Reasons to accept:
- Very well-written paper (with beautiful plots) that addresses an under-studied question: the importance of specific samples seen during fine-tuning at test time in the multilingual and cross-lingual settings.
- Many settings and dimensions of the problem are analyzed and discussed, such as sample importance in the multilingual setting, in the zero-shot cross-lingual setting, and the impact of fine-tuning dynamics on cross-lingual data sharing.

Reasons to Reject:
- None

Suggestion:
- Given the emergence of LLMs as an essential framework in NLP, discussing how this method could be applied to pretrained LLMs (without task-specific fine-tuning) and their multilingual generalization would be highly beneficial for the paper.

---

### Decision · Program_Chairs · 2023-10-07

**Decision:**

Accept-Main

**Comment:**

In the context of multilingual masked language models, this paper focuses on understanding the importance of specific training (i.e. fine-tuning) data samples at test time (referred to as cross-lingual data sharing). More specifically, it measures and analyzes the importance of the training samples and their language distribution using TracIn (Prudi et al. 2020) for the multilingual and cross-lingual performance of XLM-R across three tasks.

Reasons to accept:
- Very well-written paper (with beautiful plots) that addresses an under-studied question: the importance of specific samples seen during fine-tuning at test time in the multilingual and cross-lingual settings.
- Many settings and dimensions of the problem are analyzed and discussed, such as sample importance in the multilingual setting, in the zero-shot cross-lingual setting, and the impact of fine-tuning dynamics on cross-lingual data sharing.

Reasons to Reject:
- None

Suggestion:
- Given the emergence of LLMs as an essential framework in NLP, discussing how this method could be applied to pretrained LLMs (without task-specific fine-tuning) and their multilingual generalization would be highly beneficial for the paper.